# Accelerating Diffusion Planners in Offline RL via Reward-Aware Consistency Trajectory Distillation

**Xintong Duan**[*]  **Yutong He**[*]  **Fahim Tajwar**  **Ruslan Salakhutdinov**
**J. Zico Kolter**  **Jeff Schneider**
Carnegie Mellon University
{xintongd,yutonghe,ftajwar,rsalakhu,zkolter,jeff4}@cs.cmu.edu

## Abstract

Although diffusion models have achieved strong results in decision-making tasks, their slow inference speed remains a key limitation. While consistency models offer a potential solution, existing applications to decision-making either struggle with suboptimal demonstrations under behavior cloning or rely on complex concurrent training of multiple networks under the actor-critic framework. In this work, we propose a novel approach to consistency distillation for offline reinforcement learning that directly incorporates reward optimization into the distillation process. Our method achieves single-step sampling while generating higher-reward action trajectories through decoupled training and noise-free reward signals. Empirical evaluations on the Gym MuJoCo, FrankaKitchen, and long horizon planning benchmarks demonstrate that our approach can achieve a $9.7\%$ improvement over previous state-of-the-art while offering up to $142\times$ speedup over diffusion counterparts in inference time.

## 1 Introduction

Recent advances in diffusion models have demonstrated their remarkable capabilities across various domains (Song et al., 2020a; Karras et al., 2022; Liu et al., 2023; Chi et al., 2023; Janner et al., 2022), including decision-making tasks in reinforcement learning (RL). These models excel particularly in capturing multi-modal behavior patterns (Janner et al., 2022; Chi et al., 2023; Ajay et al., 2022) and achieving strong out-of-distribution generalization (Duan et al., 2025; Block et al., 2023), making them powerful tools for complex decision-making scenarios. However, their practical deployment faces a significant challenge: the computational overhead of the iterative sampling procedures, which requires numerous denoising steps to generate high-quality outputs.

To address this limitation, various diffusion acceleration techniques have been proposed, including ordinary or stochastic differential equations (ODE or SDE) solvers with flexible step sizes (Song et al., 2020a; Lu et al., 2022; Karras et al., 2022), sampling step distillation (Song et al., 2023; Kim et al., 2023) and improved noise schedules and parametrizations (Salimans & Ho, 2022; Song & Dhariwal, 2024). In particular, consistency distillation (Song et al., 2023) and consistency trajectory models (Kim et al., 2023) have emerged as one of the most promising solutions for image generation, in which a many-step diffusion model serves as a teacher to train a student consistency model that achieves comparable performance while enabling faster sampling through a single-step or few-step generation process.

This breakthrough has sparked considerable interest in applying consistency-based distillation to decision-making tasks. However, current applications either adopt a behavior cloning approach (Lu et al., 2024; Prasad et al., 2024; Wang et al., 2024) or integrate few-step diffusion based samplers in actor-critic frameworks (Chen et al., 2023; Ding & Jin, 2023; Li et al., 2024). While promising, these approaches face inherent challenges: behavior cloning performs well only with expert demonstrations but struggles with suboptimal data (e.g., median-quality replay buffers), while actor-critic methods

---

[*]Equal contribution

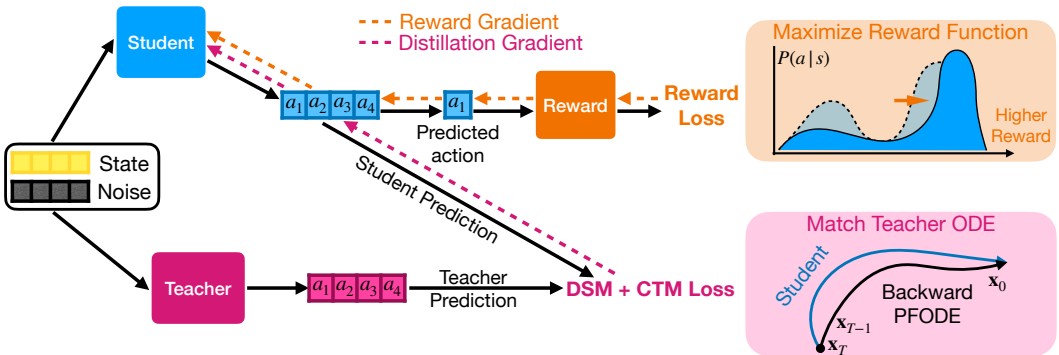

Figure 1: Overview of Reward Aware Consistency Trajectory Distillation (RACTD). We incorporate reward guidance with consistency trajectory distillation to train a student model that can generate actions with high rewards with only one denoising step.

require concurrent optimization of multiple networks from scratch with sensitive hyperparameters, leading to training complexity, instability, and computational overhead.

This raises an important question: can we develop a more effective approach to consistency-based distillation that tackles the two fundamental challenges introduced in previous framework: suboptimal data and the concurrent training of actor critic networks for offline RL? We address this challenge by introducing a novel method that directly incorporates reward optimization into the consistency trajectory distillation process. Our approach builds on existing infrastructure, leveraging a pre-trained diffusion planner as teacher and a standalone reward model to augment the standard consistency trajectory distillation (Kim et al., 2023) with an explicit reward objective for a single-step sampling student. The vanilla consistency trajectory distillation helps the student planner distill the diverse behavior modes learned by the teacher from mixed-quality offline data into a single-step sampling model. The additional reward objective acts as a mode selection mechanism: effectively steering the student planner toward selecting high reward modes from the multi-modal distributions captured by the teacher diffusion planner.

Our decoupled training framework offers two additional key advantages: training simplicity and flexibility. By generating clean action trajectories through single-step diffusion sampling, our approach removes the need for noise-aware reward models required for classifier-guided sampling from multi-step diffusion models (Janner et al., 2022), which also suffer from reduced accuracy at high noise levels. Furthermore, the reward model is trained independently from the teacher diffusion planner and the distillation process, avoiding the tight coupling that arises from concurrent multi-network training present in actor-critic methods. By integrating reward optimization directly into the single-step distillation process, our method simultaneously enables efficient generation and achieves superior performance, while maintaining straightforward training procedures.

We demonstrate the performance and sampling efficiency of our RACTD on the suboptimal hetero-geneous D4RL Gym-MuJoCo and D4RL FrankaKitchen benchmark and challenging long-horizon planning task Maze2d (Fu et al., 2020). Our method demonstrates both superior performance and substantial sampling efficiency compared to existing approaches, achieving an $9.7\%$ improvement compared to existing state-of-the-art (SOTA) while maintaining a $142$-fold reduction in sampling time inherited from the consistency trajectory distillation.

Our contributions include: (1) We propose a novel reward-aware consistency trajectory distillation method for offline RL that enables single-step action trajectory generation while achieving superior performance, (2) We demonstrate that our approach enables decoupled training without the com-plexity of concurrent multi-network optimization or noise aware reward model training, (3) Through comprehensive experiments on multi-modal suboptimal dataset and long-horizon planning tasks, we show that our method achieves $9.7\%$ improvement over prior SOTA while achieving up to $142\times$ speedup by leveraging consistency trajectory distillation. We will publicly release our code upon acceptance of this paper.

## 2 BACKGROUND

### 2.1 PROBLEM SETTING

In this paper we consider the classic setting of offline reinforcement learning, where the goal is to learn a policy $\pi$ to generate actions that maximize the expected cumulative discounted reward in a Markov decision process (MDP). A MDP is defined by the tuple $(\mathcal{S}, \mathcal{A}, \mathcal{P}, R, \gamma)$, where $\mathcal{S}$ is the set of possible states $\boldsymbol{s} \in \mathcal{S}$, $\mathcal{A}$ is the set of actions $\boldsymbol{a} \in \mathcal{A}$, $\mathcal{P}(\boldsymbol{s}' \mid \boldsymbol{s}, \boldsymbol{a})$ is the transition dynamics, $R(\boldsymbol{s}, \boldsymbol{a})$ is a reward function, and $\gamma \in [0, 1]$ is a discount factor. In offline RL, we further assume that the agent can no longer interact with the environment and is restricted to learning from a size $M$ static dataset $\mathcal{D} = \{\boldsymbol{\tau}_i\}_{i=1}^{M}$, where $\boldsymbol{\tau} = (\boldsymbol{s}_0, \boldsymbol{a}_0, r_0, \boldsymbol{s}_1, \boldsymbol{a}_1, r_1, \ldots, \boldsymbol{s}_H, \boldsymbol{a}_H, r_H)$ represents a rollout of episode horizon $H$ collected by following a behavior policy $\pi_\beta$.

Mathematically, we want to find a policy $\pi^*$ that

$$\pi^* = \arg\max_{\pi} \mathbb{E}_{\boldsymbol{\tau} \sim \pi} \left[ \sum_{n=0}^{H} \gamma^n R(\boldsymbol{s}_n, \boldsymbol{a}_n) \right] \tag{1}$$

subject to the constraint that all policy evaluation and improvement must rely on $\mathcal{D}$ alone.

### 2.2 DIFFUSION MODELS

Diffusion models generate data by learning to reverse a gradual noise corruption process applied to training examples. Given a clean data sample $\boldsymbol{x}_0$, we define $\boldsymbol{x}_t$ for $t \in [0, T]$ as increasingly noisy versions of $\boldsymbol{x}_0$. The forward (or noising) process is commonly formulated as an Itô stochastic differential equation (SDE):

$$\mathrm{d}\boldsymbol{x} = f(\boldsymbol{x}, t)\mathrm{d}t + g(t)\mathrm{d}w \tag{2}$$

where $w$ is a standard Wiener process. As $t$ approaches the final timestep $T$, the distribution of $\boldsymbol{x}_T$ converges to a known prior distribution, typically Gaussian. At inference time, the model reverses this corruption process by following the corresponding reverse-time SDE, which depends on the score function $\nabla_{\boldsymbol{x}} \log p_t(\boldsymbol{x})$. In practice, this score function is approximated by a denoiser network $D_\phi$, enabling iterative denoising from $\boldsymbol{x}_T$ back to $\boldsymbol{x}_0$. An alternative, deterministic interpretation of the reverse process is given by the probability flow ODE (PFODE):

$$\mathrm{d}\boldsymbol{x} = \left[ f(\boldsymbol{x}, t) - \tfrac{1}{2} g(t)^2 \nabla_{\boldsymbol{x}} \log p_t(\boldsymbol{x}) \right] \mathrm{d}t \tag{3}$$

which preserves the same marginal distribution $p_t(\boldsymbol{x})$ as the reverse SDE at each timestep $t$. This ODE formulation often enables more efficient sampling through larger or adaptive step sizes without significantly compromising the sample quality.

EDM (Karras et al., 2022) refine both the forward and reverse processes through improved noise parameterization and training objectives. Concretely, they reparametrize the denoising score matching (DSM) loss so that the denoiser network learns to predict a scaled version of the clean data:

$$\mathcal{L}_{\mathrm{EDM}} = \mathbb{E}_{t, \boldsymbol{x}_0, \boldsymbol{x}_t | \boldsymbol{x}_0} \left[ d(\boldsymbol{x}_0, D_\phi(\boldsymbol{x}_t, t)) \right] \tag{4}$$

where $d$ is a distance metric in the clean data space. In this paper, we train an EDM model as the teacher using the pseudo huber loss as $d$ following Prasad et al. (2024). At inference time, EDM solves the associated PFODE with a 2nd-order Heun solver.

### 2.3 CONSISTENCY TRAJECTORY DISTILLATION

The iterative nature of the diffusion sampling process introduces significant computational overhead. Among various acceleration techniques proposed, consistency distillation (Song et al., 2023) has emerged as a particularly effective approach. The core idea is to train a student model that can emulate the many-step denoising process of a teacher diffusion model in a single step.

Building upon this framework, Kim et al. (2023) introduced Consistency Trajectory Models (CTM). Instead of learning only the end-to-end mapping from noise to clean samples, CTM learns to predict across arbitrary time intervals in the diffusion process. Specifically, given three arbitrary timesteps $0 \leq k < u < t \leq T$, CTM aims to align two different paths to predict $\boldsymbol{x}_k$: (1) direct prediction from time $t$ to $k$ using the student model, and (2) a two-stage prediction that first uses a numerical solver (e.g., Heun) with the teacher model to predict from time $t$ to $u$, and then uses the student model to predict from time $u$ to $k$.

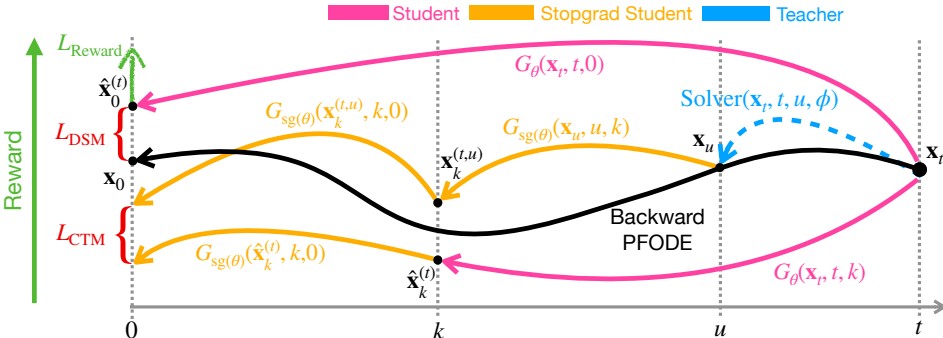

Figure 2: Visualization of CTM loss, DSM loss and reward loss.

Since the distance metric $d$ is defined on the clean data space and may not be well-defined in the noisy data space, in practice we further map all predictions to time 0 using the student model. Formally, denote the student model as $G_\theta$, the CTM loss is defined as:

$$\mathcal{L}_{\text{CTM}} = \mathbb{E}\left[d\left(G_{sg(\theta)}(\hat{\boldsymbol{x}}_k^{(t)}, k, 0), G_{sg(\theta)}(\boldsymbol{x}_k^{(t,u)}, k, 0)\right)\right] \tag{5}$$

where $G_\theta(\boldsymbol{x}_t, t, u)$ represents the student prediction from time $t$ to $u$ given noisy sample $\boldsymbol{x}_t$ at time $t$, $\text{sg}(\theta)$ represents stop-gradient student parameters and

$$\hat{\boldsymbol{x}}_k^{(t)} = G_\theta(\boldsymbol{x}_t, t, k), \qquad \boldsymbol{x}_k^{(t,u)} = G_{\text{sg}(\theta)}(\text{Solver}(\boldsymbol{x}_t, t, u; \phi), u, k) \tag{6}$$

Here $\text{Solver}(\boldsymbol{x}_t, t, u; \phi)$ is the numerical solver result from time $t$ to $u$ using the teacher model $D_\phi$ given noisy sample $\boldsymbol{x}_t$ at time $t$.

In addition to the CTM loss, Kim et al. (2023) also incorporates the DSM loss to enforce the generated samples to be close to the training data. The DSM loss for the student model is the same as the one for EDM in Equation 4:

$$\mathcal{L}_{\text{DSM}} = \mathbb{E}_{t, \boldsymbol{x}_0, \boldsymbol{x}_t | \boldsymbol{x}_0}\left[d(\boldsymbol{x}_0, G_\theta(\boldsymbol{x}_t, t, 0))\right] \tag{7}$$

CTM loss together with DSM loss constitutes consistency trajectory distillation (CTD), resembling the model in Kim et al. (2023) without the optimal GAN loss. In Figure 2 we provide a visualization of these objectives on a PFODE trajectory. After the distillation, the student model can perform "anytime-to-anytime" jumps along the PFODE trajectory. One-step sampling can then be achieved by calculating $\hat{\boldsymbol{x}}_0^{(T)} = G_\theta(\boldsymbol{x}_T, T, 0)$. This one-step sampling ability unlocks significant speedups and many potential design choices that are beneficial in RL, which we will elaborate next section.

## 3 METHOD

### 3.1 MOTIVATION AND INTUITION

Diffusion models and their consistency-based counterparts have demonstrated promising results in decision-making tasks, particularly in capturing multimodal behavior patterns (Janner et al., 2022; Chi et al., 2023; Ajay et al., 2022). Common recipes for using these models in decision-making generally fall into one of the three paradigms: (1) training a diffusion or consistency planner on expert demonstrations via behavior cloning and deploying it directly as a policy (Chi et al., 2023; Prasad et al., 2024); (2) integrating diffusion or consistency models into actor-critic frameworks for RL tasks (Wang et al., 2022; Hansen-Estruch et al., 2023; Ding & Jin, 2023); or (3) using reward agnostic diffusion planner with guided diffusion sampling for RL tasks (Janner et al., 2022; Ajay et al., 2022).

While behavior cloning pipelines perform well when trained on expert demonstrations, they often struggle with suboptimal datasets (e.g., medium-replay buffers) collected from diverse behavior policies. Such datasets typically exhibit complex multimodal behavior patterns, where only some modes lead to high rewards. Although one could potentially use rejection sampling to filter out low-reward training data, this approach becomes prohibitively sample inefficient, particularly as the quality of the training data deteriorates. On the other hand, to generate high reward actions, actor-critic

approaches require concurrent training of multiple neural networks with sensitive hyperparameters. Finally, guided diffusion sampling necessitates training noise-aware reward models and multi-step sampling, which could be detrimental for time-sensitive decision-making tasks like self-driving.

So how can we better leverage potentially suboptimal datasets to design a diffusion-based single-step sampling model with a simple training procedure? Our key idea is to utilize the multimodal information captured by the teacher diffusion planner and encourage the student diffusion planner to favor the high reward modes. We achieve this by incorporating a reward objective directly in the consistency distillation process. Since our student model can achieve single-step denoising, we can incorporate a reward model trained in the clean sample space and avoid the multi-step reward optimization for diffusion models.

### 3.2 MODELING ACTION SEQUENCES

When applying diffusion models to RL, several modeling choices are available: modeling actions (as a policy), modeling rollouts (as a planner), or modeling state transitions (as a world model). Following Chi et al. (2023), we adopt a planner approach and model a fixed-length sequence of future actions conditioned on a fixed-length sequence of observed states. This formulation ensures consecutive actions form coherent sequences, and reduces the chances of generating idle actions.

Formally, let $\vec{s}_n = \{s_{n-h}, s_{n-h+1}, \ldots, s_n\}$ denote a length-$h$ sequence of past states at rollout time $n$, and $\vec{a}_n = \{a_n, a_{n+1}, \ldots, a_{n+c}\}$ represent a length-$c$ sequence of future actions. Both the teacher and the student learn to model the conditional distribution $p(\vec{a}_n \mid \vec{s}_n)$. In the context of diffusion notations, $x = \vec{a}_n \mid \vec{s}_n$.

During execution, we can either execute only the first predicted action $a_n$ in the environment before replanning at the next step, or follow the entire predicted sequence of actions at once.

### 3.3 REWARD-AWARE CONSISTENCY TRAJECTORY DISTILLATION

Based on our action trajectory formulation, we now present our approach to integrating reward optimization into the consistency trajectory distillation process.

Let $R_\psi$ be a pre-trained differentiable return-to-go network (i.e. reward model) that takes the state $s_n$ and action $a_n$ at rollout time $n$ as inputs and predicts the future discounted cumulative reward $\hat{r}_n = \sum_{j=0}^{H-n} \gamma^j r_{n+j}$. We implement this reward model with four ConvBlocks used by previous reward-guided sampling method (Janner et al., 2022), followed by a Linear layer to map output to the correct dimension (Appendix B). When the student model generates a prediction $\vec{a}_n \mid \vec{s}_n = \hat{x}_0^{(T)} = G_\theta(x_T, T, 0)$, we extract the action at time $n$, denoted as $\hat{a}_n$, from the predicted sequence and pass it along with $s_n$ to the frozen reward model $R_\psi$ to estimate $\hat{r}_n$. The goal of our reward-aware training is to maximize the estimated discounted cumulative reward. Mathematically, the reward objective is defined as

$$\mathcal{L}_{\text{Reward}} = -R_\psi(\vec{s}_n, \hat{a}_n) \tag{8}$$

The final loss for reward-aware consistency trajectory distillation (RACTD) combines all three terms:

$$\mathcal{L} = \alpha \mathcal{L}_{\text{CTM}} + \beta \mathcal{L}_{\text{DSM}} + \sigma \mathcal{L}_{\text{Reward}} \tag{9}$$

where $\alpha$, $\beta$, and $\sigma$ are hyperparameters to balance different loss terms. This objective bears resemblance to established offline RL algorithms like off-policy deterministic policy gradient (Silver et al., 2014) (Appendix G), suggesting our approach builds upon well recognized theoretical foundations.

### 3.4 DECOUPLED TRAINING

A key advantage of our method of combining reinforcement learning, diffusion models, and consistency distillation is the ability to support fully decoupled training of all components. Traditional actor-critic frameworks, which are commonly used to incorporate diffusion models into reinforcement learning, require simultaneous training of both the actor and critic networks from scratch. This concurrent optimization presents considerable challenges, often demanding extensive hyperparameter tuning and careful balancing of different learning objectives.

Guided diffusion sampling, as proposed in Janner et al. (2022), offers an alternative approach by taking inspiration from classifier guided diffusion (Dhariwal & Nichol, 2021; Song et al., 2020b). However, since these classifiers (i.e. reward models) must evaluate noisy states at every denoising step

to provide gradient guidance, they require noise-aware training and thus cannot be separately trained from the diffusion model. Also, predicting the correct reward from highly corrupted input could be very challenging, which can lead to inaccurate guidance that accumulates during its multi-step sampling process.

Our method, in contrast, fully leverages the advantages of single-step denoising models by operating entirely in the noise-free state-action space. This design choice enables the reward model to provide stable and effective signals without requiring noise-aware training. Importantly, the reward model can be pre-trained completely decoupled from the teacher model and distillation process. This separation not only simplifies the training process but also allows for flexible integration of different reward models using the same teacher model.

### 3.5 REWARD OBJECTIVE AS MODE SELECTION

In offline RL, models often have to learn from datasets containing behaviors of varying quality. While diffusion models excel at capturing these diverse behavioral modes, they inherently lack the ability to differentiate between actions that lead to high versus low rewards. Our RACTD addresses this limitation by transforming the reward-agnostic teacher diffusion sampling distribution into one that preferentially samples from high-reward modes. We empirically verify this through a comparative analysis using the D4RL hopper-medium-expert dataset Fu et al. (2020), which contains an equal mixture of expert demonstrations and suboptimal rollouts from a partially trained policy.

Figure 3 illustrates the reward distributions of rollouts sampled from three models: the unconditioned teacher, unconditioned student, and RACTD. The dataset (grey) exhibits two distinct modes corresponding to medium-quality and expert rollouts. The unconditioned teacher model (blue) accurately captures this bimodal distribution, and the unconditioned student model (orange) faithfully replicates it. In contrast, our RACTD (green) concentrates its samples on the higher-reward mode, demonstrating that our reward guidance effectively identifies and selects optimal behaviors from the teacher's multi-modal distribution. We also include the discussion between sample diversity and mode selection in Appendix F.

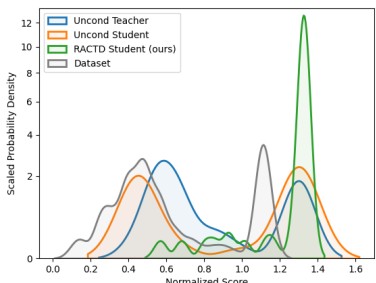

Figure 3: The reward distribution of the D4RL hopper-medium-expert dataset and 100 rollouts from an unconditioned teacher, an unconditioned student, and RACTD.

## 4 EXPERIMENT

In this section, we conduct experiments to demonstrate: (1) the effectness of our RACTD in identifying high reward behavior patterns among multimodal mix quality data, (2) the expressiveness of our single step model compared to its multistep counterpart on complex high dimensional long horizon planning tasks, and (3) the speed-up achieved over the teacher model and existing policy-based diffusion models inherited from our appropriate application of CTD.

### 4.1 OFFLINE RL

**Baselines** We compare our approach against a comprehensive set of baselines, including vanilla behavior cloning (BC) and Consistency Policy (Consistency BC) (Ding & Jin, 2023); model-free RL algorithms CQL (Kumar et al., 2020) and IQL (Kostrikov et al., 2021); model-based algorithms Trajectory Transformer (TT) (Janner et al., 2021), MOPO (Yu et al., 2020), MOReL (Kidambi et al., 2020), MBOP (Argenson & Dulac-Arnold, 2020); autoregressive model Decision Transformer (DT) Chen et al. (2021); diffusion-based planner Diffuser Janner et al. (2022); and diffusion/flow-based actor-critic methods Diffusion QL (Wang et al., 2022), Consistency AC (Ding & Jin, 2023), and Flow Q-learning (Park et al., 2025).

**Setup** We evaluate our method and the baselines on D4RL Gym-MuJoCo and FrankaKitchen benchmark (Fu et al., 2020). Gym-MuJoCo is a popular offline RL benchmark that contains three environments (hopper, walker2d, halfcheetah) of mixtures of varying quality data (medium-replay, medium, medium-expert). The FrankaKitchen dataset features a 9-DoF robotic arm performing multi-task manipulation across four kitchen objects (microwave, kettle, light, cabinets) with mixed-quality demonstrations (partial, mixed).

Following the setups in the prior works, we report the performance of both online and offline model selection if available in their original paper. Online model selection reports the best-performing checkpoint observed during training, while offline model selection reports the performance of the last training epoch. Results for non-diffusion-based models and Diffuser are taken from Janner et al. (2022), and results for Diffusion QL and Consistency AC/BC are sourced from their respective papers. The results for RACTD are reported as the mean and standard error over 100 planning seeds. We use past observation length $h = 1$ and prediction horizon $c = 16$ and closed-loop planning in all experiments following the same setup in previous diffusion planners (Chi et al., 2023; Prasad et al., 2024) and include the ablations on observation and planning horizons in Appendix E.6. The best score is emphasized in bold and the second-best is underlined.

Table 1: (Offline RL: Gym-MuJoCo) Performance and sampling efficiency (NFE: Number of Function Evaluations) of RACTD and a variety of baselines on the D4RL Gym-MuJoCo benchmark.

| | Medium Expert | | | Medium | | | Medium Replay | | | Avg ↑ | NFE ↓ |
|---|---|---|---|---|---|---|---|---|---|---|---|
| | HalfCheetah | Hopper | Walker2d | HalfCheetah | Hopper | Walker2d | HalfCheetah | Hopper | Walker2d | | |
| **Offline model selection** | | | | | | | | | | | |
| BC | 55.2 | 52.5 | 107.5 | 42.6 | 52.9 | 75.3 | 36.6 | 18.1 | 26.0 | 51.9 | - |
| CQL | 91.6 | 105.4 | 108.8 | 44.0 | 58.5 | 72.5 | 45.5 | 95.0 | 77.2 | 77.6 | - |
| IQL | 86.7 | 91.5 | 109.6 | 47.4 | 66.3 | 78.3 | 44.2 | 94.7 | 73.9 | 77.0 | - |
| DT | 86.8 | 107.6 | 108.1 | 42.6 | 67.6 | 74.0 | 36.6 | 82.7 | 66.6 | 74.7 | - |
| TT | 95.0 | 110.0 | 101.9 | 46.9 | 61.1 | 79.0 | 41.9 | 91.5 | 82.6 | 78.9 | - |
| MOPO | 63.3 | 23.7 | 44.6 | 42.3 | 28.0 | 17.8 | 53.1 | 67.5 | 39.0 | 42.1 | - |
| MOReL | 53.3 | 108.7 | 95.6 | 42.1 | 95.4 | 77.8 | 40.2 | 93.6 | 49.8 | 72.9 | - |
| MBOP | **105.9** | 55.1 | 70.2 | 44.6 | 48.8 | 41.0 | 42.3 | 12.4 | 9.7 | 47.8 | - |
| Diffusion QL | 96.8 ±0.3 | 111.1 ±1.3 | 110.1 ±0.3 | 51.1 ±0.5 | 90.5 ±4.6 | 87.0 ±0.9 | 47.8 ±0.3 | 101.3 ±0.6 | 95.5 ±1.5 | 87.9 | 5 |
| Consistency AC | 84.3 ±4.1 | 100.4 ±3.5 | 110.4 ±0.7 | **69.1** ±0.7 | 80.7 ±10.5 | 83.1 ±0.3 | **58.7** ±3.9 | 99.7 ±0.5 | 79.5 ±3.6 | 85.1 | 2 |
| Consistency BC | 32.7 ±1.2 | 90.6 ±9.3 | 110.4 ±0.7 | 31.0 ±0.4 | 71.7 ±8.0 | 83.1 ±0.3 | 34.4 ±5.3 | 99.7 ±0.5 | 73.3 ±5.7 | 69.7 | 2 |
| Diffuser | 88.9 ±0.3 | 103.3 ±1.3 | 106.9 ±0.2 | 42.8 ±0.4 | 74.3 ±1.4 | 79.6 ±0.55 | 37.7 ±0.5 | 93.6 ±0.4 | 70.6 ±1.6 | 77.5 | 20 |
| RACTD(Ours) | 88.5 ±2.1 | **120.2** ±2.6 | **122.3** ±0.3 | 56.6 ±0.6 | **102.4** ±2.4 | **112.8** ±1.3 | 51.4 ±0.2 | **108.3** ±1.1 | **105.2** ±1.8 | **96.4** | **1** |
| **Online model selection** | | | | | | | | | | | |
| Diffusion QL | 97.2 ±0.4 | 112.3 ±0.8 | 111.2 ±0.9 | 51.5 ±0.3 | 96.6 ±3.4 | 87.3 ±0.5 | 48.3 ±0.2 | 102.0 ±0.4 | 98.0 ±0.5 | 89.3 | 5 |
| Consistency AC | 89.2 ±3.3 | 106.0 ±1.3 | 111.6 ±0.7 | **71.9** ±0.8 | 99.7 ±2.3 | 84.1 ±0.3 | **62.7** ±0.6 | 100.4 ±0.6 | 83.0 ±1.5 | 89.8 | 2 |
| Consistency BC | 39.6 ±3.4 | 96.8 ±4.6 | 111.6 ±0.7 | 46.2 ±0.4 | 78.3 ±2.6 | 84.1 ±0.3 | 45.4 ±0.7 | 100.4 ±0.6 | 80.8 ±3.4 | 75.9 | 2 |
| RACTD(Ours) | 95.9 ±1.5 | **129.0** ±1.3 | **122.3** ±0.3 | 59.3 ±0.2 | **115.5** ±1.7 | **118.8** ±0.3 | 57.9 ±1.0 | **109.5** ±0.3 | **105.2** ±1.8 | **101.5** | **1** |

**Results** As shown in Table 1, RACTD achieves the highest average score by a substantial margin and best or second-best performance on $8/9$ tasks in Gym-MuJoCo, with the only exception being a medium-expert dataset where reward guidance is less beneficial. RACTD is also the only planning-based method that achieves single-step sampling compared to consistency model based actor-critic methods (which require double sampling steps) and other diffusion-based planners (which require $20\times$ more sampling steps).

As shown in Table 2, RACTD consistently outperforms its counterpart that incorporates an actor-critic framework for consistency models (Consistency AC) or flow-based models (Flow q-learning) across all dataset qualities. On suboptimal partial and mixed datasets where reward guidance is crucial, our method achieves near-equivalent performance to multi-step diffusion approaches with $5\times$ fewer sampling steps. These findings highlight the effectiveness of our reward-aware distillation framework, which combines decoupled training, noise-free reward modeling, and single-step sampling efficiency.

Table 2: (Offline RL: FrankaKitchen) Performance and sampling efficiency (NFE) of RACTD and diffusion based baselines on D4RL FrankaKitchen benchmark. Each cell has two values: the first is offline model selection and the second (in brackets) is online model selection.

| Method | kitchen-partial | kitchen-mixed | Avg↑ | NFE↓ |
|---|---|---|---|---|
| Diffusion QL | **60.5** ±6.9 (63.7 ±5.2) | **62.6** ±5.1 (66.6 ±3.3) | **61.6**(65.15) | 5 |
| Consistency AC | 38.2 ±1.8 (39.8 ±1.6) | 45.8 ±1.5 (46.7 ±0.9) | 42.0(43.3) | 2 |
| Consistency BC | 22.6 ±3.8 (23.8 ±2.8) | 23.5 ±1.8 (24.3 ±1.3) | 23.1(24.1) | 2 |
| Flow Q-learning | 47.5 ±2.8 (53.3 ±3.2) | 46.0 ±2.2 (53.5 ±1.9) | 46.8(53.4) | 1 |
| RACTD(Ours) | 59.0 ±1.5 (63.1 ±0.1) | 60.9 ±0.6 (61.9 ±0.2) | 60.0(62.5) | 1 |

## 4.2 LONG HORIZON PLANNING

Next, we showcase the expressiveness of the single-step sampling model RACTD in complex high-dimensional long-horizon planning tasks. Previously, Diffuser has shown great potential in open-loop long-horizon planning, but requires a significantly larger number of denoising steps compared to closed-loop planning like MuJoCo. We demonstrate that our model can achieve superior performance with a single-step denoising process under the same problem formulation.

**Setup** We test this ability on D4RL Maze2d (Fu et al., 2020), which is a sparse reward long-horizon planning task where an agent may take hundreds of steps to reach the goal in static environments. Maze2d medium and large are also shared environments in other benchmarks like OGBench (Park et al., 2024), where it is demonstrated to be even more challenging than higher-dimensional environments like AntMaze. Following the setup in Janner et al. (2022), we use a planning horizon $128, 256, 384$ for U-Maze, Medium and Large respectively, all conditioned on two observations: the start and goal position. We perform open-loop planning by generating the entire state sequence followed by a reverse dynamics model to infer all the actions from the predicted state sequence. The reward model returns $1$ if the current state reaches the goal and $0$ otherwise. The baseline results are reported from Janner et al. (2022) and RACTD results are reported as the mean and standard error of $100$ planning seeds.

**Results** As shown in Table 3, both Diffuser and RACTD outperform previous model-free RL algorithms CQL, IQL, Flow Q-learning, and MPPI. Our approach surpasses the Diffuser baseline in all settings, highlighting its ability to effectively capture complex behavioral patterns and high-dimensional information in the training dataset. Notably, the planning dimension for this task (384 for the Large Maze) is substantially higher than that of MuJoCo tasks (16). As a result, Diffuser requires significantly more denoising steps (256 for the Large Maze) compared to MuJoCo (20 steps). On the contrary, despite the increased task complexity, RACTD still only requires a single denoising step to achieve $11.6\times$ performance boost, showcasing its expressiveness. Furthermore, maintaining the speedup inherited from CTD, our RACTD is able to achieve significant performance improvement over CTD, especially on the more challenging long-horizon tasks.

Table 3: (Long-horizon planning) The performance of RACTD, Diffuser, Flow Q-learning and prior model-free algorithms in the Maze2D environment. Flow Q-learning is close loop planning and Diffuser, RACTD are open loop planning.

| Method | U-Maze | | | Medium | | | Large | | | Avg Score |
|---|---|---|---|---|---|---|---|---|---|---|
| | Score | NFE | Time (s) | Score | NFE | Time (s) | Score | NFE | Time (s) | |
| MPPI | 33.2 | - | - | 10.2 | - | - | 5.1 | - | - | 16.2 |
| CQL | 5.7 | - | - | 5 | - | - | 12.5 | - | - | 7.7 |
| IQL | 47.4 | - | - | 34.9 | - | - | 58.6 | - | - | 47.0 |
| Flow Q-learning | 106.7 ±6.4 | **1** | - | 89.7 ±1.9 | **1** | - | 107.3 ±7.1 | **1** | - | 101.2 |
| Diffuser | 113.9 ±3.1 | 64 | 1.664 | 121.5 ±2.7 | 256 | 4.312 | 123.0 ±6.4 | 256 | 5.568 | 119.5 |
| CTD | 123.4 ±1.0 | **1** | **0.029** | 119.8 ±4.1 | **1** | **0.047** | 127.1 ±6.4 | **1** | **0.049** | 123.4 |
| RACTD (Ours) | **125.7** ±0.6 | **1** | **0.029** | **130.8** ±1.8 | **1** | **0.047** | **143.8** ±0.0 | **1** | **0.049** | **133.4** |

## 4.3 INFERENCE TIME COMPARISON

Beyond performance improvements, thanks to the single-step sampling ability from consistency trajectory distillation, our RACTD significantly accelerates diffusion-based models while maintaining their superior performance for decision-making tasks. The primary computational bottleneck in diffusion models arises from the multiple function evaluations (NFEs) required by the iterative denoising process. By reducing the number of denoising steps to a single NFE, our approach achieves a speed-up roughly proportional to the number of denoising steps originally required.

**Setup** To evaluate sampling efficiency, we compare RACTD with different samplers, including DDPM (Ho et al., 2020), DDIM (Song et al., 2020a), and EDM (Karras et al., 2022) using the same network architecture as our teacher and student model. Additionally, we report the efficiency of Diffuser. Note that since Diffuser employs a different model architecture and generates future state-action pairs, its sampling time may also be influenced by these factors. Table 4 and Table 3 present the wall clock sampling time and NFE for MuJoCo (hopper-medium-replay) and Maze2d. All experiments are conducted on one NVIDIA Tesla V100-SXM2-32GB.

**Results** In hopper-medium-replay, RACTD achieved $20\times$ reduction in NFEs and a $43\times$ speed-up compared to Diffuser. Additionally, our student model requires $80\times$ fewer NFEs and samples $142\times$ faster than the teacher model. In Maze2d, RACTD significantly accelerates computation compared to Diffuser, achieving approximately $57\times$, $92\times$, and $114\times$ speed-ups on Umaze, Medium and Large mazes, by reducing NFEs by a factor of 256 for the Medium and Large mazes.

Table 4: Wall clock time and NFEs per action for different samplers and Diffuser on MuJoCo hopper-medium-replay.

| Method | Time (s) | NFE | Score |
|---|---|---|---|
| Diffuser | 0.644 | 20 | 93.6 |
| DDPM | 0.236 | 15 | 24.2 |
| DDIM | 0.208 | 15 | 60.6 |
| EDM (Teacher) | 2.134 | 80 | **114.2** |
| RACTD (Ours, Student) | **0.015** | **1** | 109.5 |

Table 5: We compare incorporating the reward model in different stages of training on MuJoCo hopper-medium-replay. Results are presented as the mean and standard error across 100 seeds.

| Hopper Medium-Replay | Unconditioned teacher | Reward-Aware teacher |
|---|---|---|
| Unconditioned student | 50.8 ±0.3 | 96.2 ±0.2 |
| Reward-Aware student | **109.5** ±0.3 | 96.0 ±0.3 |

## 5 ABLATION STUDY

We evaluate the necessity of incorporating reward objectives during student versus teacher training and include extended analyses on reward loss weights and multi-step sampling in Appendix E.3 E.5.

### 5.1 IMPACT OF REWARD OBJECTIVE

To understand the unique advantages of incorporating reward objectives during distillation, we conduct a systematic comparison across four model configurations: the baseline combination of an unconditioned teacher and student, a reward-aware teacher paired with an unconditioned student, a fully reward-aware teacher-student pair, and our proposed RACTD, which combines an unconditioned teacher with a reward-aware student. The results for hopper-medium-replay and walker-medium is shown in Table 5 and Table 12.

Our analysis reveals that while incorporating reward objectives at any stage yields substantial improvements, optimal performance is achieved through our RACTD framework, which combines an unconditioned teacher with reward-aware student distillation. Our method allows the teacher to capture a comprehensive range of behavioral patterns, while enabling the student to selectively distill the most effective strategies. Although incorporating reward objectives in the teacher model also enhances performance, this approach risks discarding suboptimal behaviors that may be valuable in novel testing scenarios, potentially limiting the model's generalization capabilities.

## 6 RELATED WORK

**Diffusion Models in Reinforcement Learning** Diffusion models have emerged as a powerful approach for decision-making tasks in reinforcement learning Janner et al. (2022); Ajay et al. (2022); Wang et al. (2022); Hansen-Estruch et al. (2023); Chi et al. (2023). The integration of diffusion models into RL frameworks typically follows two main paradigms: actor-critic approaches, where diffusion models serve as policy or value networks (Wang et al., 2022; Hansen-Estruch et al., 2023), and policy-based approaches, where diffusion models directly generate action trajectories Janner et al. (2022); Ajay et al. (2022); Chi et al. (2023). While these methods have demonstrated impressive performance on standard RL benchmarks, their practical deployment is hindered by the slow sampling time inherent to vanilla diffusion policies based on DDPM Ho et al. (2020). This limitation poses particular challenges for speed-sensitive real-world applications such as robotics.

**Accelerating Diffusion Model Sampling** Various approaches have been proposed to accelerate the sampling process in diffusion models. One prominent direction leverages advanced ODE solvers to reduce the number of required denoising steps (Song et al., 2020a; Karras et al., 2022; Lu et al., 2022). Another line of work explores knowledge distillation techniques Luhman & Luhman (2021); Salimans & Ho (2022); Berthelot et al. (2023); Kim et al. (2023); Song et al. (2023), where student models learn to take larger steps along the ODE trajectory. Particularly, consistency trajectory models (Kim et al., 2023) enable one-step sampling by learning anytime-to-anytime jumps along the PFODE trajectory, achieving performance that surpasses teacher models.

**Consistency Models in Decision Making** Consistency models have emerged as a promising policy class for behavior cloning from expert demonstrations in robotics (Lu et al., 2024; Prasad et al., 2024; Wang et al., 2024). In RL, several works have enhanced actor-critic methods by replacing traditional diffusion-based value/policy networks with consistency models, showing faster inference and training speed (Chen et al., 2023; Ding & Jin, 2023; Li et al., 2024). These approaches directly

incorporate consistency loss (Song et al., 2023) into the value/policy network training, rather than distilling a separate student model. Park et al. (2025) distilled a flow-based model into a single-step student but still requires concurrent network optimization under the actor critic framework. While Wang et al. made some initial attempts to apply consistency distillation in policy-based RL through classifier-free guidance and reverse dynamics, their approach requires two NFEs and under-performs the state-of-the-art. In contrast, RACTD is a straightforward approach of using a separate reward model and incorporating reward objective during student distillation, achieving superior performance with only one NFE.

## 7 CONCLUSION

In this work, we address the challenge of accelerating diffusion planners at sampling by introducing reward-aware consistency trajectory distillation (RACTD), which predicts high-reward action trajectories in a single denoising step. RACTD uses a pre-trained diffusion teacher planner and a separately trained reward model, leveraging the teacher's ability to capture multi-modal distributions while prioritizing higher-reward modes to generate high-quality action trajectories from suboptimal training data. Its decoupled training approach avoids the complex concurrent optimization of multiple networks and enables the use of a standalone, noise-free reward model. RACTD outperforms previous state-of-the-art by $9.7\%$ while leveraging consistency trajectory distillation to accelerate its diffusion counterparts up to a factor of $142$.

## IMPACT STATEMENT

This paper presents work whose goal is to advance the field of Machine Learning. There are many potential societal consequences of our work, none which we feel must be specifically highlighted here.

## ACKNOWLEDGMENT

This work was partly supported by DSTA, ONR N000142312368 and ONR MURI N00014-25-1-2116. FT was partially supported by the U.S. Army Futures Command under Contract No. W519TC-23-C-0030 during the project.

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

## A    MODEL ARCHITECTURE

We follow the model architecture used in Chi et al. (2023) and Prasad et al. (2024) and continue to use the 1D temporal CNN layer for our Unet and FiLM layers to process the conditioning information.

### A.1    MODEL SIZES FOR MAZE2D

Model parameters for teacher models of Umaze, Medium, and Large Maze are shown below in Table 6. The student model has the same architecture as the teacher model except it also takes one extra variable of denoising timestep as conditioning.

| Parameter | Umaze | Medium | Large |
|---|---|---|---|
| diffusion_step_embed_dim | 256 | 256 | 256 |
| down dims | [256, 512, 1024] | [512, 1024, 2048] | [256, 512, 1024, 2048] |
| horizon | 128 | 256 | 384 |
| kernel size | 5 | 5 | 5 |

Table 6: Model parameters for Unet in Maze2d.

### A.2    MODEL SIZES FOR GYM-MUJOCO

Unet parameters for teacher and student models in the MuJoCo task are shown below in Table 7. Model sizes are fixed through 9 different environments.

| Parameter | MuJoCo |
|---|---|
| diffusion_step_embed_dim | 128 |
| down dims | [512, 1024, 2048] |
| horizon | 16 |
| kernel size | 5 |

Table 7: Model parameters for Unet in MuJoCo.

## B    REWARD MODEL PARAMETERS

We follow the setup in Janner et al. (2022), where we use Linear layers and Mish layers Misra (2019) for the reward model. We stack 4 ConvBlocks for the reward model, each with dimension shown in Table 14. Each ConvBlock consists of a Conv1D layer, groupnorm, and a Mish layer. Reward model architecture remains the same across all MuJoCo benchmarks.

## C    TRAINING DETAILS

Our models are trained on D4RL dataset (Fu et al., 2020), which was released under Apache-2.0 license.

### C.1    NOISE SCHEDULER

We follow the setup in Prasad et al. (2024) and use EDM noise scheduler (Karras et al., 2022) for the teacher model. Particularly, discretization bins are chosen to be 80.

Student model used CTM scheduler (Kim et al., 2023) also with discretization bins of 80.

### C.2    WEIGHT OF DIFFERENT LOSSES

The weights for CTM, DSM, and Reward loss we used in the experiment are shown below in Table 9. Generally, if the training dataset includes more expert samples, the weight for reward guidance is

| Parameter | MuJoCo |
|---|---|
| layer dimensions | [32, 64, 128, 256] |

Table 8: Reward model parameters in MuJoCo.

| Parameter | CTM | DSM | Reward |
|---|---|---|---|
| hopper-medium-replay | 1.0 | 1.0 | 0.8 |
| hopper-medium | 1.0 | 1.0 | 3.0 |
| hopper-medium-expert | 1.0 | 1.0 | 0.0 |
| walker2d-medium-replay | 1.0 | 1.0 | 1.0 |
| walker2d-medium | 1.0 | 1.0 | 0.4 |
| walker2d-medium-expert | 1.0 | 1.0 | 0.2 |
| halfcheetah-medium-replay | 1.0 | 1.0 | 1.0 |
| halfcheetah-medium | 1.0 | 1.0 | 0.5 |
| halfcheetah-medium-expert | 1.0 | 1.0 | 0.0 |

Table 9: Weights for CTM, DSM, and Reward loss used in MuJoCo benchmark.

smaller. A reward weight of 0.0 resembles behavior cloning with consistency trajectory distillation. We found that as long as the loss weights are chosen to keep the individual loss terms within the same order of magnitude, the model will achieve reasonable performance.

# D MORE RESULTS

## D.1 EXPERT AND RANDOM DATASET PERFORMANCE FOR GYM-MUJOCO

We include the performance on expert and random datasets along with a comparison between dataset quality, ranging from random to expert, with another diffusion-based planning method Diffuser (Janner et al., 2022). Under the trajectory modeling setup, the random dataset provides little valid trajectory and the expert dataset will degrade the problem to behavior cloning. Nevertheless, our method outperforms Diffuser in all settings tested.

| hopper | random | medium-replay | medium | medium-expert | expert | NFE |
|---|---|---|---|---|---|---|
| RACTD | 31.8±0.0 | 104.9±2.1 | 121.0±0.5 | 129.0±1.3 | 137.1±0.1 | 1 |
| Diffuser | 6.7±0.1 | 70.6±1.6 | 79.6±0.6 | 106.9±0.2 | 110.3±0.1 | 20 |

Table 10: Performance of Diffuser and RACTD on Gym-MuJoCo random and expert dataset.

## D.2 EXPERT DATASET PERFORMANCE FOR FRANKA KITCHEN

Since kitchen-complete consists of pure expert demonstrations, the problem degrades to behavior cloning and reward guidance becomes less effective.

Table 11: (Offline RL: FrankaKitchen) Performance and sampling efficiency (NFE) of RACTD and diffusion based baselines on FrankaKitchen complete. Each cell has two values: the first is offline model selection and the second (in brackets) is online model selection.

| Method | kitchen-complete | NFE↓ |
|---|---|---|
| Diffusion QL | **84.0** ±7.4 (84.5 ±6.1) | 5 |
| Consistency AC | 51.9 ±6.0 (67.6 ±2.7) | 2 |
| Consistency BC | 45.2 ±5.0 (50.9 ±3.6) | 2 |
| RACTD(Ours) | 56.3 ±8.2 (58.1 ±8.3) | **1** |

# E MORE ABLATIONS

## E.1 REWARD OBJECTIVE FOR WALKER-MEDIUM

Table 12: We compare incorporating the reward model in different stages of training on MuJoCo walker-medium. Results are presented as the mean and standard error across 100 seeds.

| Walker Medium | Unconditioned teacher | Reward-Aware teacher |
|---|---|---|
| Unconditioned student | 93.3 ±1.8 | 97.0 ±1.0 |
| Reward-Aware student | **118.8** ±0.3 | 94.5 ±2.6 |

## E.2 COMPARING WITH FAST SAMPLING ALGORITHMS FOR MAZE2D

Table 13: We compare fast sampling algorithms DDIM and CTD, along with our method RACTD on Maze2d environment. DDIM performs fast sampling based on a DDPM model, while CTD and RACTD (ours) distill an EDM teacher. The number of function evaluations (NFE) reflects the sampling speed of each algorithm. Results are reported as the mean and standard error over 100 random seeds.

| Method | NFE | U-Maze Score | Medium Score | Large Score | Average Score |
|---|---|---|---|---|---|
| DDPM | 100 | **126.3**±0.7 | 126.8 ±3.0 | 144.8 ±4.9 | 132.6 |
| DDIM | 10 | 121.2 ±1.1 | 126.2 ±2.8 | 143.1 ±4.9 | 130.2 |
| DDIM | **1** | 3.5 ±4.7 | −2.6 ±12.5 | −1.5 ±0.7 | −0.2 |
| EDM | 80 | 125.4 ±0.6 | 120.1 ±4.2 | **149.0** ±0.5 | 131.5 |
| CTD | 1 | 123.4 ±1.0 | 119.8 ±4.1 | 127.1 ±6.4 | 123.4 |
| RACTD (ours) | 1 | 125.7 ±0.6 | **130.8** ±1.8 | 143.8 ±0.0 | **133.4** |

## E.3 EFFECT OF REWARD OBJECTIVE WEIGHTS

The reward loss weight is a hyperparameter in our pipeline that impacts both training stability and performance. We plot the reward curve achieved over 200 epochs of student training with reward loss weights ranging from $[0.3, 0.8, 1.5]$ on MuJoCo hopper-medium-replay in Figure 4. The mean and standard error are reported across 20 rollouts from intermediate checkpoints.

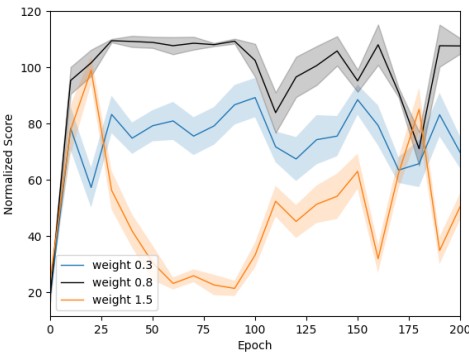

Figure 4: Ablation on reward objective weight.

With lower weights, increasing the weight leads to higher performance and relatively stable training. However, when the weight is too high (e.g. 1.5 in this plot), evaluation initially increases but fluctuates

as training progresses. This occurs when the reward loss dominates DSM and CTM losses, resulting in unstable training.

### E.4 DSM AND REWARD LOSS WEIGHTS

We include the ablation for different DSM and reward loss weights. The other two losses are fixed at optimal when altering the target loss.

| DSM weight | 0.0 | 0.3 | 0.6 | 0.8 | 1.0 | 1.2 | 1.5 | 2.0 |
|---|---|---|---|---|---|---|---|---|
| **RACTD** | $1.5 \pm 0.0$ | $23.8 \pm 1.0$ | $35.6 \pm 2.2$ | $105.2 \pm 1.8$ | $\mathbf{108.3} \pm 1.1$ | $100.0 \pm 2.3$ | $104.9 \pm 1.5$ | $93.6 \pm 3.4$ |

Table 14: DSM loss weight ablation for RACTD in MuJoCo.

| Reward weight | 0.0 | 0.3 | 0.5 | 0.7 | 0.8 | 0.9 | 1.0 | 1.5 | 2.0 |
|---|---|---|---|---|---|---|---|---|---|
| **RACTD** | $50.8 \pm 0.3$ | $69.8 \pm 2.5$ | $91.5 \pm 2.5$ | $\mathbf{108.4} \pm 1.4$ | $108.3 \pm 1.1$ | $85.4 \pm 3.2$ | $100.9 \pm 2.6$ | $50.5 \pm 2.1$ | $17.0 \pm 0.0$ |

Table 15: Reward loss weight ablation for RACTD in MuJoCo.

### E.5 NUMBER OF SAMPLING STEPS

Since our student model is trained for anytime-to-anytime jumps, it naturally extends to multi-step denoising without additional training. Following the approach in Song et al. (2023), given intermediate denoising timesteps $0 < t_1 < t_2 < T$, we first denoise from $T$ to 0 as usual. We then add noise again to $t_1$ and denoise it back to 0, and repeat this process for $t_2$. This iterative refinement can enhance generation quality. We evaluate the student using 2, 3, and 4 denoising steps as reported in Table 16. Results show that multi-step sampling barely improves model performance.

Table 16: Inference time, NFE, and score comparison for student model multi-step sampling on MuJoCo hopper-medium-replay.

| NFE | Time(s) | Score |
|---|---|---|
| 1 | **0.0147** | $109.5 \pm 0.3$ |
| 2 | 0.0241 | $\mathbf{109.8} \pm 0.9$ |
| 3 | 0.0377 | $108.7 \pm 0.1$ |
| 4 | 0.0517 | $107.9 \pm 1.6$ |

### E.6 OBSERVATION AND PLANNING HORIZON

We compare the performance of incorporating different steps of past observations and predicting different numbers of future actions for MuJoCo walker medium-replay here in Table 17. We can see that the performance can be further improved by conditioning on a larger number of observations and a longer planning horizon.

Table 17: We compare the performance of using different numbers of past observations (h) and future actions (c) on MuJoCo walker medium-replay. Results are presented as the mean and standard error across 100 seeds.

| walker medium-replay | Future actions c=16 | Future actions c=8 |
|---|---|---|
| Past observation h=1 | $105.2 \pm 1.8$ | $104.3 \pm 1.4$ |
| Past observation h=4 | $\mathbf{108.0} \pm 1.4$ | $103.3 \pm 2.2$ |

## F THE TRADE-OFF BETWEEN MODE SELECTION AND SAMPLE DIVERSITY

In this section, we include a discussion about the trade-off between mode selection induced by our reward-aware training and the sample diversity of the student. Naturally, favoring selected modes can

led to generation with limited sample diversity as summarized in Table 18. This trade-off between sample diversity and sample optimality observed in RACTD is similar to what has been seen in other generative domains (e.g., language model RLHF (Huang et al., 2024), classifier-guided diffusion, conditional image generation), where preference alignment also often reduces sample diversity. In our case, the reward model acts similarly to a classifier or an alignment reward model, guiding the model toward desirable behaviors and sacrificing some of the sample diversity by design.

Importantly, our decoupled framework allows the use of a single, unconditioned teacher with strong generalization capabilities across tasks. For multi-task or unseen-task settings, different reward models can be trained per task, and corresponding student models can be distilled from the same teacher using different reward models.

Table 18: A summarization of the trade off between sample diversity and model performance.

|  | Sample diversity | Performance | Sample time |
|---|---|---|---|
| Reward agnostic diffusion | High | Low | Slow |
| Reward aware diffusion | Low | High | Slow |
| Reward agnostic consistency distillation | High | Low | Fast |
| Reward aware consistency distillation | Low | High | Fast |

## G  THEORETICAL INSIGHTS

Here we provide a proof sketch of how our reward-aware consistency trajectory distillation training resembles off-policy deterministic policy gradient (Silver et al., 2014), thus sharing the well-studied theoretical analysis of these methods for offline RL.

According to CTM (Kim et al., 2023) (Section 3), our student model generates a single-step prediction by

$$G_\theta(\boldsymbol{x}_T, T, 0) = g_\theta(\boldsymbol{x}_T, T, 0) = \boldsymbol{x}_T + \int_T^0 \frac{\boldsymbol{x}_u - \mathbb{E}[\boldsymbol{x}|\boldsymbol{x}_u]}{u} du$$

From Taylor expansion, we have $G_\theta(\boldsymbol{x}_T, T, 0) = \mathbb{E}[\boldsymbol{x}_0|\boldsymbol{x}_T] + \mathcal{O}(T)$

Let $\mu_\theta = \mathbb{E}[\boldsymbol{x}_0|\boldsymbol{x}_T]$. Then the gradient of our reward model is

$$\nabla_\theta R(G_\theta(\boldsymbol{x}_T, T, 0)) = \nabla_\theta R(\mu_\theta) \nabla_\theta \mathbb{E}[\boldsymbol{x}_0|\boldsymbol{x}_T] = \nabla_\theta R(\mu_\theta) \nabla_\theta \mu_\theta$$

For off-policy deterministic policy gradient[cite], we have

$$\nabla_\theta J_\beta(\mu_\theta) \approx \mathop{\mathbb{E}}_{\boldsymbol{s} \sim \rho^\beta} [\nabla_\theta \mu_\theta(\boldsymbol{s}) \nabla_{\boldsymbol{a}} Q^\mu(\boldsymbol{s}, \boldsymbol{a})|_{\boldsymbol{a} = \mu_\theta(\boldsymbol{s})}]$$

, where $\rho^\beta$ is the behavior policy.

$$\nabla_\theta J_\beta(\mu_\theta) \approx \mathop{\mathbb{E}}_{s \sim \rho^\beta} [\nabla_\theta \mu_\theta(s) \nabla_a Q^\mu(s, a)|_{a = \mu_\theta(s)}]$$

Our method first samples a state $\boldsymbol{s}$ from the offline dataset collected by the behavior policy $\rho^\beta$ and asks the student model to predict actions $\boldsymbol{a}$ given $\boldsymbol{s}$. Then, the reward model takes in $\boldsymbol{s}$ and $\boldsymbol{a}$ and predicts the corresponding Q value. Therefore $R(\mu_\theta)$ is equivalent to $Q^\mu(\boldsymbol{s}, \boldsymbol{a}|_{\boldsymbol{s} \sim \rho^\beta, \boldsymbol{a} = \mu_\theta(\boldsymbol{s})})$. Thus, our reward-aware student model training resembles off-policy deterministic policy gradient.

## H  LIMITATIONS AND FUTURE WORK

One limitation of our approach is the need to train three separate networks: the teacher, student, and reward model. Training the teacher can be time-consuming, as achieving strong performance often

requires a higher number of denoising steps. Additionally, consistency trajectory distillation is prone to loss fluctuations, and incorporating a reward model into the distillation process may further amplify this instability. Future work will focus on developing a more stable and efficient training procedure, as well as exploring methods to integrate non-differentiable reward models into the framework.

