# OpenReview forum: "Accelerating Diffusion Planners in Offline RL via Reward-Aware Consistency Trajectory Distillation"
_ICLR.cc/2026/Conference — ICLR 2026 Poster_

### Official Review · Reviewer_ixyB · 2025-10-29

**Soundness:** 3
**Presentation:** 3
**Contribution:** 3
**Rating:** 6
**Confidence:** 4

**Summary:**

This paper introduces Reward-Aware Consistency Trajectory Distillation (RACTD), a method that extends consistency trajectory models to the offline reinforcement learning setting by incorporating a reward objective into the distillation process. The approach allows a student model to generate high-reward action trajectories in a single denoising step, combining efficiency from consistency models with reward-guided behavior selection. Experiments across standard offline RL benchmarks demonstrate that RACTD improves policy quality and sampling efficiency compared to diffusion-based and actor-critic baselines by a large margin in many cases.

**Strengths:**

- The paper is very well written and was genuinely an enjoyable read. Most questions that arise while reading are quickly addressed by the text, and all of the necessary context to follow the technical discussion is included.
- The motivation is strong, with clear explanations of the limitations of prior work such as the slow inference speed of diffusion models and the sensitivity of actor-critic frameworks to hyperparameters.
- The introduction of decoupled training in Section 3.4 is a genuine strength, making the method simpler and more stable in practice. The idea of operating directly in the clean reward space rather than training noise-aware reward models is both elegant and effective.
- The proposed method demonstrates substantial empirical gains over prior work, with strong improvements in performance and efficiency.
- The inference-time acceleration that this method inherits from prior work makes the approach especially appealing for real-world decision-making tasks.

**Weaknesses:**

- Not a substantial weakness, but there is a typo between lines 202 and 203 (struggle vs. struggles)
- In Table 3, the paper introduces CTD as a comparison point without tying this acronym to a particular method and without comparing CTD's results to anything else in the main text (unless I am mistaken). Presumably this is Consistency Trajectory Distillation. Reading back through the paper, it appears that CTD corresponds to the authors’ approach without the reward aware component, that is, Consistency Trajectory Model (CTM) [1] augmented with a denoising score matching (DSM) loss. However, this relationship is never made explicit. Clarifying whether CTD is simply CTM+DSM loss applied to offline reinforcement learning or a modified variant beyond the addition of the DSM loss is important for readers to properly interpret the results.
- Based on the prior point, the novelty of the proposed method is therefore somewhat unclear when compared to the backbone it builds off of. The paper introduces Reward Aware Consistency Trajectory Distillation (RACTD), which extends CTD by incorporating a reward term. This alongside the incorporation of the DSM loss is novel and shows good results compared to prior methods in the literature. So the motivation for these additions is clear, as shown in Figure 3, but the relative benefit of the reward aware RACTD component remains mostly untested against the unconditional variants CTD and CTM. A broader ablation comparing CTM, CTD, and RACTD across all benchmarks (for example, Tables 1, 2, 4, and 5), alongside their relative inference times and sample quality, would better isolate the benefit of the reward awareness and DSM loss integration.
- If CTD is effectively a reimplementation of CTM adapted to the offline RL setting with minimal modifications, then the faster inference times should be credited to prior work rather than presented as new contributions. This is backed by Table 3's comparison of CTD to RACTD where both methods have identical inference times. Applying existing techniques in new settings is a contribution that is noteworthy and part of this work, but the main contribution here would then lie in the reward aware adaptation, DSM loss, and application in a new domain with great score improvement results, rather than claiming the inference speedups already established by CTM. The acceleration results therefore represent a successful domain adaptation of a pre-existing backbone you built on top of, not a new methodological innovation, and should be framed as such in the paper (e.g. the leading sentence in Section 4.3 "Beyond performance improvements, another major contribution of our work is significantly accelerating diffusion-based models for decision-making tasks." or the last sentence of the conclusion: "RACTD outperforms previous state-of-the-art by 9.7% while accelerating its diffusion counterparts up to a factor of 142.").
- Given that the primary acceleration mechanism is inherited from Consistency Trajectory Models [1], the title “Accelerating Diffusion Planners in Offline RL via Reward-Aware Consistency Trajectory Distillation” may overstate the novelty of the acceleration component. A more accurate framing would emphasize the reward-aware extension rather than the inference-speed improvement itself, which seemingly follows directly from prior work.

[1] Kim, Dongjun, et al. "Consistency trajectory models: Learning probability flow ode trajectory of diffusion." arXiv preprint arXiv:2310.02279 (2023).

**Questions:**

- In Section 3.3 the reward model is frozen during training. How sensitive is the overall planner to the diversity and coverage of the dataset used to train this reward model? The paper emphasizes robustness to suboptimal data, but what happens when planned trajectories drift outside the dataset’s support, potentially leading to compounding errors in both the reward model and the planner?
- How sensitive is your method to alpha, beta, and sigma in the RACTD loss? Do you perform ablations to determine this sensitivity?
- In Figure 3, the diffusion-based methods appear to have one of their high-probability density modes lying outside the support of the high-reward mode in the training dataset. Am I misinterpreting this figure? Intuitively, the highest-probability regions for the unconditional diffusion-based methods should align with the high-reward mode of the training distribution. The rightward shift of the RACTD distribution is understandable due to the added reward objective, but why do the unconditional diffusion-based models also shift toward higher rewards than the dataset’s high-reward mode?

---

> ### Author Response · Authors · 2025-11-21
>
> We thank the reviewer for their constructive feedback and positive assessment of our work, especially regarding our paper quality, strong motivation, elegant and effective decoupled training, substantial empirical gains, and inference-time acceleration. We would like to address your concerns below.
>
> 1. **Clarification on “CTD”:** We apologize for the confusion. CTD represents the consistency trajectory model distilled with CTM loss + DSM loss, as introduced in Kim et al 2023, without the optional GAN loss in the original paper. We use the term “CTD” as an abbreviation for “consistency trajectory distillation”, in order to distinguish it from the broader generative model family “consistency trajectory model”, which can be trained from scratch without a teacher model. In other words, CTD is our application of a variant of CTM to offline RL and we do not wish to claim it as a novelty of our work. We thank the reviewer for pointing this out and have added the corresponding clarification in the background.
>
> 2. **Novelty being unclear** We would like to summarize our key contribution as identifying **where** and **how** consistency-based distillation should interface with reward signals and policy updates in RL:
>
> - **Where:** During student distillation only (not teacher training or concurrent critic optimization).
> - **How:** Via direct reward training loss on clean single-step outputs (not noise-aware guidance or actor-critic optimization).
>
> This design is non-trivial, as prior attempts to incorporate consistency models with RL face various challenges: Without reward signals (e.g., Consistency Policy and Consistency BC), the model cannot distinguish high-reward actions and performs poorly (Tables 1-3, 5). On the other hand, actor-critic-based methods (e.g., Consistency AC) require concurrent multi-network optimization and sensitive hyperparameters, and still achieve suboptimal results (Tables 1-2).
>
> Our approach provides three benefits that prior work cannot simultaneously achieve: **single-step efficiency**, **decoupled training**, and **state-of-the-art performance**. Thus, RACTD provides a simpler yet more effective offline RL formulation than existing consistency/flow-based actor-critic methods.
>
> 3. **Ablation on loss weighting:** Thank you for this suggestion. We have added an additional ablation study on the loss weighting with respect to the DSM loss and the reward loss below on the MuJoCo hopper-medium-replay task.
>
> | DSM weight | RACTD           |
> |-----------:|-----------------|
> | 0.0        | 1.5 ± 0.0       |
> | 0.3        | 23.8 ± 1.0      |
> | 0.6        | 35.6 ± 2.2      |
> | 0.8        | 105.2 ± 1.8     |
> | 1.0        | 108.3 ± 1.1     |
> | 1.2        | 100.0 ± 2.3     |
> | 1.5        | 104.9 ± 1.5     |
> | 2.0        | 93.6 ± 3.4      |
>
> | Reward weight | RACTD          |
> |--------------:|----------------|
> | 0.0           | 50.8 ± 0.3     |
> | 0.3           | 69.8 ± 2.5     |
> | 0.5           | 91.5 ± 2.5     |
> | 0.7           | 108.4 ± 1.4    |
> | 0.8           | 108.3 ± 1.1    |
> | 0.9           | 85.4 ± 3.2     |
> | 1.0           | 100.9 ± 2.6    |
> | 1.5           | 50.5 ± 2.1     |
> | 2.0           | 17.0 ± 0.0     |
>
> As we can observe, both DSM loss and the reward loss are crucial components for the success of RACTD. When the weight for DSM loss is 0, the objective resembles CTM loss + reward loss, while the reward loss weight 0 represents vanilla CTD. We have also included the inference time and performance comparisons between CTD and RACTD for Maze2d in Table 3.
>
> 4. **Better framing of our claims:** We thank the reviewer for raising this point. We have modified the opening sentence of Section 4.3 to: "Beyond performance improvements, another major contribution of our work is significantly accelerating diffusion-based models *while maintaining their superior performance* for decision-making tasks". Regarding the title, we emphasize "via Reward-Aware Consistency Trajectory Distillation" to indicate that the reward-aware adaptation is the key methodological contribution, while the acceleration comes from the appropriate application of CTM to RL. We are happy to modify the title if clearer framing would better represent the work's contributions.
>
> 5. **Sensitivity to data diversity and coverage:** Our reward model is trained with the same dataset as the planner model for each task. In our Appendix D.1, we conducted an ablation study to show RACTD’s performance across datasets of varying quality and coverage from random (low quality but high diversity) to expert (high quality but low diversity). While the model performs poorly on purely random data, the performance from medium-replay to expert roughly remains in a strong range, indicating the model’s robustness to suboptimal data and highlighting the effectiveness of reward maximization via mode selection.

---

> ### Author Response · Authors · 2025-11-21
>
> 6. **Tajectories outside dataset support:** We thank the reviewer for raising this question and would like to clarify that our robustness claim is primarily about the suboptimal data’s portion (e.g., 100% or 50% random) and quality (e.g., collected by fully random or half-trained behavior policies), but less about generalizing to unsupported data. Also, the reward model in RACTD is only used during training to shape the student. At test time, the planner acts without querying the reward model, so we do not accumulate reward-model errors during rollout. We agree that being able to handle unsupported data and compounding error is a key challenge in offline RL, and it will be an interesting direction for future work.
>
>
> 7. **Shift in teacher distribution towards higher reward regions:** Thank you for this insightful question. The shift in teacher distribution comes from our design choice of action sequence modeling in combination with closed-loop execution. When generating a length-$c$ action sequence, the model can implicitly filter out actions that can lead to early failures (i.e. rollout termination). During closed-loop planning, we replan and only execute the first predicted action at each RL timestep. When repeated, this strategy stitches high-reward actions at different timesteps in one rollout, thus generating rollouts that yield higher rewards than the ones in the dataset. This observation is supported by our ablation study in Appendix E.6, where we show that elongating the planning horizon can effectively improve the performance.

---

> > ### Comment · Reviewer_ixyB · 2025-11-21
> >
> > Thank you for the clarifications and the additional analysis. I appreciate the effort to address the earlier concerns.
> >
> > My overall view is unchanged. The technical ideas are interesting and the empirical performance is strong. I am not opposed to acceptance. However, my main concern about the contribution framing remains unresolved, and I would strongly caution against accepting the paper without properly addressing the acceleration framing. The paper continues to present acceleration as a core contribution, despite the fact that the inference speed is inherited directly from CTM. This creates a disconnect between the stated contribution and the admitted source of the speedup. A clearer separation between inherited benefits and novel components would greatly strengthen the paper.
> >
> > The revision to Section 4.3 does not change the underlying issue. It still presents acceleration as something achieved by the proposed method rather than as an inherited property of CTM. This is not a matter of wording, but a matter of accurately attributing contributions. I understand the inclination to include acceleration as a contribution since this method is faster than prior methods, but I must reiterate that your contributions did not cause this acceleration. They are a byproduct of the chosen backbone. Applying existing techniques in new settings can absolutely be a contribution, as I noted in my original review, but the framing should reflect that distinction. As written, the framing is misleading. Other examples persist, such as the last sentences of both the abstract and the conclusion, which continue to present acceleration as a central contribution. Leaving readers with the 142x speedup figure without attributing it to CTM results in misattribution via omission. The examples I provide are illustrative rather than exhaustive. I would expect the authors to review the entire text rather than address only the specific instance I cited. I leave the title to the authors, but given these concerns, removing the acceleration framing would provide a clearer description of the actual contribution.
> >
> > In addition, the novelty question is still difficult to fully assess. The paper would benefit from broader comparisons between CTM, CTD, and RACTD across the main benchmarks, not just on a single task. The added ablation helps but does not fully isolate the benefit of the reward aware extension relative to the underlying consistency model. I recognize that running these full comparisons may not be feasible during the rebuttal period, but as noted in my original review, this ablation across all tasks is important for properly framing the contribution. In the final version, including these comparisons in Table 1, in addition to the Maze2D comparison in Table 3, would give readers a much clearer picture of where the gains come from.
> >
> > Given these points, my score remains the same. The method is promising and the results are strong, but the framing should be more precise and the novelty relative to prior consistency based methods should be demonstrated more clearly.

---

> > > ### Author Response · Authors · 2025-11-27
> > >
> > > We thank the reviewer again for the constructive feedback. We have extended the RACTD vs. CTD comparison to the full set of MuJoCo locomotion and Kitchen tasks. These expanded evaluations further support our claims that (1) RACTD provides substantial performance improvements over standard CTD especially when the data is highly suboptimal ( medium-replay cases for MuJoCo), and (2) the benefit holds across domains, including locomotion, Maze2D, and Kitchen. Please let us know if there are any questions that we can help address!
> > >
> > >
> > >
> > > **MuJoCo results:**
> > >
> > > | Model | hopper_mr    | hopper_m     | hopper_me     | walker_mr    | walker_m     | walker_me     | cheetah_mr   | cheetah_m    | cheetah_me   | Avg    |
> > > |-------|--------------|--------------|---------------|--------------|--------------|---------------|--------------|--------------|--------------|--------|
> > > | CTD   | 50.8 ± 0.3   | 58.7 ± 0.9   | 120.2 ± 2.6   | 30.5 ± 2.9   | 93.3 ± 1.8   | 117.3 ± 1.3   | 51.4 ± 0.8   | 56.5 ± 0.2   | 88.5 ± 2.1   | 74.1 |
> > > | RACTD | 108.3 ± 1.1  | 102.4 ± 2.4  | 120.2 ± 2.6   | 105.2 ± 1.8  | 112.8 ± 1.3  | 122.3 ± 0.3   | 51.4 ± 0.2   | 56.6 ± 0.6   | 88.5 ± 2.1   | 96.4 |
> > >
> > > **Kitchen results:**
> > >
> > > |  Model  | Kitchen partial                 | Kitchen mix                      | Avg |
> > > |-------|---------------------------------|----------------------------------|----|
> > > | CTD   | 52.6 ± 16.9 (59.6 ± 0.9)        | 56.7 ± 3.5 (60.0 ± 1.6)          | 54.7(59.8)    |
> > > | RACTD | 59.0 ± 1.5 (63.1 ± 0.1)         | 60.9 ± 0.6 (61.9 ± 0.2)          | 60.0(62.5)   |
> > >
> > >
> > > **Maze2d results:**
> > >
> > > | Model        | U-Maze        | Medium         | Large          | Avg    |
> > > |-------------|---------------|----------------|----------------|--------|
> > > | CTD         | 123.4 ± 1.0   | 119.8 ± 4.1    | 127.1 ± 6.4    | 123.4 |
> > > | RACTD | 125.7 ± 0.6   | 130.8 ± 1.8    | 143.8 ± 0.0    | 133.4 |

---

> ### Author Response · Authors · 2025-11-21
> **Paper revision for better CTM attribution**
>
> We sincerely thank the reviewer for their prompt and thorough feedback. We sincerely apologize that our initial rebuttal failed to adequately address the attribution concerns, and we now fully understand the issue. As a result, we have thoroughly edited the paper to properly credit CTM while clearly distinguishing our contributions. In particular, we have made the following changes (marked in red in the manuscript):
>
> 1. In the last sentence of the abstract, we have changed “while offering up to 142$\times$ speedup over diffusion counterparts in inference time” to “while **leveraging CTM (Kim et al. 2023)** to offer up to $142\times$ speedup over diffusion counterparts in inference time”.
>
> 2. In line 40-41, we have added another mention to CTM to emphasize its impact.
>
> 3. Line 54, 72 and 94, fixing typo that may cause inaccurate attribution.
>
> 4. Line 74-75, changing “The vanilla consistency trajectory distillation helps the student planner cover the diverse behavior modes learned by the teacher from mixed-quality offline data” to “The vanilla consistency trajectory distillation helps the student planner **distill** the diverse behavior modes learned by the teacher from mixed-quality offline data **into a single-step sampling model**”.
>
> 5. Line 92-93, changing “achieving an 9.7% improvement compared to existing state-of-the-art (SOTA) and a 142-fold reduction in sampling time” into “achieving an $9.7\%$ improvement compared to existing state-of-the-art (SOTA) **while maintaining** a $142$-fold reduction in sampling time **inherited from the consistency trajectory distillation**”.
>
> 6. Line 99, changing “while achieving up to 142× speedup” to “while achieving up to $142\times$ speedup **by leveraging consistency trajectory distillation**”.
>
> 7. Adding “**This one-step sampling ability unlocks significant speedups and many potentials for design choices that are beneficial in RL, which we elaborate in the next section**” to the end of Section 2.
>
> 8. Line 302-303, changing from “the speed-up achieved over the teacher model and existing policy-based diffusion models” to “the speed-up achieved over the teacher model and existing policy-based diffusion models **inherited from our appropriate application of CTD**”.
>
> 9. Adding additional discussion on the performance comparison of RACTD v.s. CTD: “**Furthermore, maintaining the speedup inherited from CTD, our RACTD is able to achieve significant performance improvement over CTD, especially on the more challenging high-dimensional tasks**”.
>
> 10. Changing the opening of Section 4.3 to “Beyond performance improvements, **thanks to the single-step sampling ability from consistency trajectory distillation, our RACTD** significantly accelerates diffusion-based models **while maintaining their superior performance** for decision-making tasks.”
>
> 11. Changing the last sentence the the conclusion from “while accelerating its diffusion counterparts up to a factor of 142” to “while **leveraging consistency trajectory distillation to accelerate** its diffusion counterparts up to a factor of $142$”
>
> We would like to emphasize that we do not wish to diminish CTM’s contribution and we sincerely thank the reviewer for pointing this problem with our previous framing. We believe that this revision clarifies our contribution and makes the distinction between our method and vanilla CTD clearer.
>
> We are committed to extending our RACTD v.s. CTD ablation to all MuJoCo tasks, although as the reviewer has mentioned it may not be feasible to complete within the rebuttal timeframe. Nevertheless, we will update the results as soon as they are available in both the rebuttal discussion and the final revision. We are also open to further revisions of the paper including the title and we welcome any additional suggestions for improving the paper.
>
> Thank you again for your constructive feedback and patience in helping us improve this work.

---

### Official Review · Reviewer_7SAP · 2025-10-31

**Soundness:** 3
**Presentation:** 3
**Contribution:** 2
**Rating:** 2
**Confidence:** 4

**Summary:**

The manuscript proposes a method termed RACTD aimed at accelerating diffusion-based components within offline reinforcement learning by reducing the number of function evaluations (NFE) while maintaining policy performance.
This work presents a novel approach to consistency distillation that directly incorporates reward optimization into the distillation process. The proposed method achieves single-step diffusion sampling while generating
higher-reward action trajectories through decoupled training and noise-free reward guidance.

**Strengths:**

- Clear experimental scope and benchmarks: Evaluation on widely used D4RL Gym-MuJoCo and FrankaKitchen datasets enhances relevance and comparability, with both offline and online model selection reported where applicable.
- Focus on sampling efficiency: Explicit reporting of NFE alongside performance indicates attention to practical efficiency, which is crucial for diffusion-based methods.

**Weaknesses:**

- Limited novelty and contribution: The core idea is to augment the distillation process with a cumulative reward maximization objective. This training pipeline has appeared in prior work (e.g., Flow Q-Learning), and the paper does not clearly isolate what is fundamentally new beyond this template.
- Central claim lacks rigorous empirical validation: The paper emphasizes incorporating a reward objective directly into consistency distillation rather than optimizing via a critic (e.g., Q-values or advantages). This design choice requires thorough ablations and head-to-head comparisons against critic-based alternatives to substantiate the claim.
- Motivation section is superficial: The insights in Section 3.1 are relatively basic and widely known; they would be more appropriate in the introduction. The main text should instead focus on deeper methodological details, theoretical justification, or non-trivial empirical findings.
- Incomplete experimental comparisons: The evaluation is missing strong, direct baselines from state-of-the-art one-step diffusion/consistency methods.

**Questions:**

The theoretical contribution appears weak; please strengthen the theoretical analysis and add comprehensive comparisons with state-of-the-art methods (e.g., one-step diffusion/consistency baselines) to substantiate the claims.

---

> ### Author Response · Authors · 2025-11-21
>
> We thank the reviewer for their constructive comments and we would like to address your concerns below.
>
> 1. **Comparison with Flow Q-Learning:** We would like to respectfully clarify that Flow Q-Learning and our methods are two fundamentally different approaches, with the following reasons:
>
>  - **Different distillation framework:** The distillation process in Flow Q-Learning is completely independent of the underlying dynamical system of diffusion/flow matching teacher. It simply applies one-step behavior cloning to a flow matching teacher's final output from multi-step sampling as a regularization to the Q-learning objective. The same procedure could be applied to any black-box policy. In particular, it does not model or exploit the time-indexed, noise-conditioned structure of diffusion/flow matching. On the other hand, our RACTD uses consistency trajectory distillation (CTD), which is specifically designed for diffusion models based on their underlying stochastic differential equations (Section 2.3). While both methods distill multi-step models into single-step models, the methodological frameworks are entirely different.
>
> - **Different ways to incorporate reward signals:** Flow Q-Learning is an actor-critic method that requires concurrent optimization of three networks: the Q function, the teacher policy and the distilled policy. Our approach on the other hand, leverages existing diffusion teachers with standalone reward models. This design choice provides several practical advantages (Sections 3.4, 3.5): it avoids the complexity and instability often associated with simultaneous actor-critic optimization, allows for modular integration with pre-trained diffusion models, and achieves superior performance compared to actor-critic baselines in our experiments.
>
> - **Different generative models:** Flow Q-Learning uses flow-matching models, whereas our method uses diffusion and consistency trajectory models.
>
> - **Different action modeling:** RACTD performs closed-loop planning by modeling action sequences while FLow Q-Learning only predicts the next action.
>
> Our approach provides three benefits that prior work cannot simultaneously achieve: **single-step efficiency**, **decoupled training**, and **state-of-the-art performance**, demonstrating RACTD provides a simpler yet more effective offline RL formulation than existing consistency/flow-based actor-critic methods.
>
> 2. **Empirical comparison with Flow Q-Learning & other actor-critic methods:** Thank you for this question. We would like to clarify that we do provide experimental comparisons with actor-critic methods in our paper. In particular, **Diffusion QL and Consistency AC** (Tables 1-2) are actor-critic methods that use diffusion/consistency models in their frameworks. Our method demonstrates superior overall performance compared to both methods in the MuJoCo environment (Table 1) and achieves comparable performance to Diffusion QL while requiring 5× fewer denoising steps in the Kitchen environment (Table 2).
>
> Following the reviewer’s suggestion, we have also added **Flow Q-Learning** in the Kitchen and Maze2d comparison. As shown in the tables below, RACTD consistently outperforms Flow Q-Learning by substantial margins in both environments, further supporting the effectiveness of our decoupled training approach over actor-critic frameworks.
>
> Maze2d Results:
> | Method          | Umaze           | Medium          | Large           | NFE |
> |-----------------|-----------------|-----------------|-----------------|:---:|
> | Flow Q-Learning | 106.67 ± 6.4    | 89.73 ± 1.9     | 107.25 ± 7.1    |  1  |
> | RACTD           | 125.7 ± 0.6     | 130.8 ± 1.8     | 143.8 ± 0.0     |  1  |
>
> Kitchen Results:
> | Method          | Partial (offline/online)    | Mixed (offline/online)       | NFE |
> |-----------------|-----------------------------|------------------------------|:---:|
> | Flow Q-Learning | 47.5 ± 2.8 (53.3 ± 3.2)     | 46.0 ± 2.2 (53.5 ± 1.9)      | 1   |
> | RACTD           | 59.0 ± 1.5 (63.1 ± 0.1)     | 60.9 ± 0.6 (61.9 ± 0.2)      | 1   |

---

> ### Author Response · Authors · 2025-11-21
>
> 3. **”Motivation section is superficial”:** We appreciate this writing feedback. Section 3.1 is intended to provide background for better understanding of the following sections and to highlight the key intuition behind the method: “Our key idea is to utilize the multimodal information captured by the teacher diffusion planner and encourage the student diffusion planner to favor the high reward modes” (lines 211-213). The remainder of the main text do cover “deeper methodological details, theoretical justification, or non-trivial empirical findings” as the reviewer suggested:
> - Methodological details: Section 3.2, 3.3
> - Theoretical justification: Section 3.3 with detailed theoretical insights in Appendix G
> - Non-trivial empirical findings: Section 3.4,3.5, and Section 4
>
> We thank the reviewers for raising this concern. We are open to any further writing suggestions and are happy to reorganize or expand any sections to improve the paper's clarity and technical depth.
>
> 4. **Theoretical contributions:** Thank you for this feedback. We provide theoretical grounding  in Appendix G by showing that RACTD follows the same update rule as off-policy deterministic policy gradient [1]. This connection demonstrates that our approach follows established principles from the policy gradient literature and provides theoretical justification for our design.
>
> However, we would like to emphasize that we position this paper as primarily an empirical work rather than a theoretical one. We demonstrate through experiments that RACTD achieves superior performance on suboptimal datasets by selecting higher-reward modes while providing significant speedup. We believe these empirical findings meaningfully advance the practical use of consistency-based models in RL.
>
> **Reference:**
>
> [1] Silver, et al. Deterministic Policy Gradient Algorithms. ICML 2014.

---

### Official Review · Reviewer_LYwS · 2025-11-01

**Soundness:** 3
**Presentation:** 3
**Contribution:** 3
**Rating:** 6
**Confidence:** 3

**Summary:**

This work introduces Reward-Aware Consistency Trajectory Distillation (RACTD), which presents a novel application of consistency distillation to diffusion-based planning in reinforcement learning. The method trains a single-step student model to emulate a multi-step diffusion teacher, effectively accelerating inference while maintaining high performance. The authors incorporate a reward model alongside the consistency trajectory loss and denoising score matching loss within a decoupled training framework. Experimental results demonstrate that the proposed RACTD substantially outperforms state-of-the-art diffusion planning methods in both performance and inference speed.

**Strengths:**

1. The paper introduces a novel application of consistency distillation, resulting in a single-step model that effectively emulates multi-step diffusion processes, thereby significantly improving inference efficiency.

2. The proposed RACTD demonstrates strong empirical performance, substantially outperforming prior state-of-the-art diffusion-based reinforcement learning methods, particularly on the D4RL benchmark.

3. The paper presents comprehensive experiments and detailed implementation descriptions, accompanied by code availability, which enhances the credibility and reproducibility of the reported results.

**Weaknesses:**

1. The idea of the paper is interesting, and the contribution of accelerating the sampling stage through consistency distillation is clear. However, the work appears to rely heavily on existing consistency distillation techniques, with limited novelty beyond their direct application to diffusion-based planning.

2. The contribution of the proposed reward-aware consistency trajectory distillation is somewhat unclear. The method appears to employ a standard reward model as an auxiliary loss applied to the student’s output, rather than introducing a fundamentally new formulation or architecture.

**Questions:**

1. The authors claim that their proposed decoupled training outperforms the simultaneous training of actor and critic networks from scratch. However, there does not appear to be any experimental evidence supporting this statement. Could the authors provide an ablation study or empirical comparison demonstrating the benefits of decoupled training?

2. In Table 2 and Table 11, Diffusion-QL achieves better performance than RACTD, although RACTD uses 5x fewer sampling steps as stated by the authors. In this case, why did the authors not compare all methods under the same number of sampling steps? Such a comparison would be more meaningful than simply showing that the baseline achieves the best performance, particularly in Table 11.

3. The results in Table 1 on the D4RL benchmarks show that the proposed RACTD method significantly outperforms prior state-of-the-art approaches on average. What do the authors believe is the main factor behind this improvement? Is the performance gain primarily attributable to the application of consistency distillation?

---

> ### Author Response · Authors · 2025-11-21
>
> We thank the reviewer for their thoughtful feedback and recognition of our work's novelty, strong empirical results, and comprehensive experiments. We now would like to address your concerns below.
> 1. **Novelty and formulation:** We would like to clarify that our contribution goes beyond simply applying consistency distillation to offline RL (which was also explored by prior work). Our key contribution is identifying **where** and **how** consistency-based distillation should interface with reward signals and policy updates for RL:
>
> - **Where:** During student distillation only (not teacher training or concurrent critic optimization).
> - **How:** Via directly optimizing reward loss on clean single-step outputs (not noise-aware classifier guidance or actor-critic optimization).
>
> This design is non-trivial, as prior attempts to incorporate consistency models with RL either **do not use reward signals** (e.g., Consistency Policy and Consistency BC), in which case the model cannot distinguish high-reward actions and performs poorly (Tables 1–3, 5), or **rely on actor–critic frameworks** (e.g., Consistency AC), which require concurrent multi-network optimization and sensitive hyperparameters yet still achieve suboptimal results (Tables 1–2).
>
> In contrast, our approach provides three benefits that prior work cannot achieve simultaneously: **single-step efficiency**, **decoupled training**, and **state-of-the-art performance**. Thus, RACTD provides a simpler yet more effective offline RL formulation fundamentally different than existing consistency/flow-based actor-critic methods, going beyond simply adding an auxiliary loss.
>
>
> 2. **Comparison to actor-critic:** Thank you for this question. We would like to clarify that we do provide experimental comparisons with actor-critic methods in our paper. In particular, **Diffusion QL and Consistency AC** (Tables 1-2) are actor-critic methods that use diffusion/consistency models in their frameworks. Our method demonstrates superior overall performance compared to both methods in the MuJoCo environment (Table 1) and achieves comparable performance to Diffusion QL while requiring 5× fewer denoising steps in the Kitchen environment (Table 2).
>
> To provide additional empirical evidence, we have added **Flow Q-learning** [1], a recent flow-based single-step actor-critic method in Kitchen and Maze2d environments. As shown below, RACTD consistently outperforms Flow Q-learning by substantial margins in both environments, further supporting the effectiveness of our decoupled training approach over actor-critic frameworks.
>
> Maze2d Results:
> | Method          | Umaze           | Medium          | Large           | NFE |
> |-----------------|-----------------|-----------------|-----------------|:---:|
> | Flow Q-learning | 106.67 ± 6.4    | 89.73 ± 1.9     | 107.25 ± 7.1    |  1  |
> | RACTD           | 125.7 ± 0.6     | 130.8 ± 1.8     | 143.8 ± 0.0     |  1  |
>
> Kitchen Results:
> | Method          | Partial (offline/online)    | Mixed (offline/online)       | NFE |
> |-----------------|-----------------------------|------------------------------|:---:|
> | Flow Q-learning | 47.5 ± 2.8 (53.3 ± 3.2)     | 46.0 ± 2.2 (53.5 ± 1.9)      | 1   |
> | RACTD           | 59.0 ± 1.5 (63.1 ± 0.1)     | 60.9 ± 0.6 (61.9 ± 0.2)      | 1   |
>
>
> 3. **Why not compare all methods with the same number of steps:** We report the results for the number of denoising steps where each model is optimized to ensure fair performance comparison. Diffusion QL is optimized for 5 steps, Consistency BC/AC for 2 steps, and Diffuser for 20 steps. As shown in our Appendix E.2 and Table 3 in [2], forcing these models to perform one-step prediction can result in catastrophic performance degradation (from score > 100 to near 0 performance). To facilitate a direct comparison with other models optimized for single-step sampling, we provide the experimental results for Flow Q-Learning shown above, where RACTD substantially outperforms across all benchmarks tested.

---

> ### Author Response · Authors · 2025-11-21
>
> 4. **Main factor behind the performance gain:** We do not attribute the performance gain to vanilla consistency distillation but to our decoupled training with noise-free reward models (Section 3.4) and high-reward mode selection setup (Section 3.5). Compared to guided sampling baselines like Diffuser, decoupled training and single-step sampling allow us to use a standalone reward model on clean samples, thus avoiding inaccurate guidance from highly noisy intermediate steps. Compared to actor–critic frameworks such as Diffusion QL, Consistency AC, and Flow Q-Learning, when the dataset is largely suboptimal, the learned Q-function is poorly constrained in the truly high-return region (with little data support), making these methods prone to exploiting spurious Q-peaks. In contrast, RACTD uses an expressive diffusion teacher to model the multimodal behavior distribution first and then uses the reward model to select high-reward modes within this learned support during distillation, which leads to more reliable and high-return behavior.
>
> Our ablation study in Table 5 (also listed below) demonstrates that vanilla CTD without reward guidance achieves only 50.8, while RACTD achieves 108.3 on Hopper-medium-replay, indicating that vanilla CTD is not the main reason behind our performance gain.
>
> | Method | Performance (Hopper-medium-replay) |
> |---------------|-------------------------|
> | vanilla CTD | 50.8 ± 0.3 |
> | RACTD | **108.3 ± 1.1** |
>
> **Reference:**
>
> [1] Park, et al. Flow Q-learning. ICML 2025.
>
> [2] Ding & Jin. Consistency Models as a Rich and Efficient Policy Class for Reinforcement Learning. ICLR 2024.

---

### Author Response · Authors · 2025-12-04

We thank the reviewers for their thoughtful feedback and would like to briefly summarize the discussion for the AC. The discussion has significantly strengthened our work and clarified our contributions.

### We appreciate the reviewers’ recognition of our paper’s strengths:

- The motivation is strong, with clear explanations of the limitations of prior work (reviewer ixyB).
- The introduction of decoupled training is a genuine strength, making the method simpler and more stable in practice. The idea of operating directly in the clean reward space rather than training noise-aware reward models is both elegant and effective (reviewer ixyB).
- There are substantial empirical gains over prior work, with strong improvements in performance and efficiency (reviewers LYwS, ixyB).
- The paper presents comprehensive experiments and detailed implementation descriptions, with clear experimental scope and benchmarks (reviewers LYwS, 7SAP).
- There is careful attention to practical efficiency, which is especially appealing for real-world decision-making tasks (reviewers 7SAP, ixyB).
- The paper is very well written and was genuinely an enjoyable read (reviewer ixyB).

---

### Key clarifications during rebuttal

**Experiments for actor–critic alternatives (reviewers LYwS, 7SAP):**  We clarified that Diffusion QL and Consistency AC are already included as actor–critic baselines (Tables 1–2), where RACTD shows superior overall performance.

**SOTA one-step diffusion/consistency baselines (reviewer 7SAP):**  We added Flow Q-Learning in the Kitchen and Maze2D environments during rebuttal. RACTD consistently outperforms Flow Q-Learning by substantial margins in both environments.

**Difference from Flow Q-Learning (reviewer 7SAP):**  We emphasized that Flow Q-Learning is an actor–critic method, while RACTD uses a different decoupled distillation framework, reward incorporation mechanism, generative model family, and action modeling.

**Novelty being unclear (reviewers LYwS, ixyB):**  We clarified that our novelty lies in identifying *where* and *how* to couple reward with consistency distillation: only in student distillation (not teacher training or concurrent critic optimization) and via a clean, single-step reward loss (not noise-aware guidance or actor-critic optimization). This yields a combination of single-step efficiency, decoupled training, and SOTA performance that prior methods do not simultaneously achieve.

**Different number of denoising steps (reviewer LYwS):**  We explained that each baseline is evaluated at its optimized number of denoising steps to ensure fair comparison, and that forcing them to one step leads to catastrophic degradation (as documented in Appendix E.2 and prior work).

**Source of performance gains (reviewer LYwS):**  We clarified that improvements are not due to vanilla CTD, but to decoupled training with a noise-free reward model (Sec. 3.4) and high-reward mode selection (Sec. 3.5).

**More ablation on reward weights and RACTD vs. CTD (reviewer ixyB):**  We extended RACTD vs. CTD comparisons to all MuJoCo and Kitchen tasks (Maze2d already included) and added more detailed ablations on DSM and reward loss weights.

**Better framing of our contributions (reviewer ixyB):**  We thoroughly edited the paper to explicitly credit CTM for single-step acceleration while clearly distinguishing our own contributions (changes marked in red in the manuscript).

**Writing, theory, and data coverage (reviewers 7SAP, ixyB):**  We clarified writing-related concerns by pointing to the corresponding paragraphs (reviewer 7SAP), directed readers to Appendix G for theoretical grounding while emphasizing that the paper is primarily empirical (reviewer 7SAP), and used Appendix D.1 to analyze performance across datasets with varying quality and coverage, from random (low quality, high diversity) to expert (high quality, lower diversity) (reviewer ixyB).

---

To summarize, RACTD achieves single-step inference while improving performance on multiple offline RL tasks. We believe these clarifications and additions resolve the main methodological and framing concerns and present RACTD as a clearly positioned, empirically strong, and practically useful approach for efficient offline RL.

---

### Meta-Review · Area_Chair_U4fq · 2026-01-11

**Summary:**

Reviewers agreed that the paper is empirically strong and addresses an important bottleneck: slow inference in diffusion-based planners. They particularly praised the decoupled training design and the focus on practical efficiency (LYwS, ixyB, 7SAP). The main decision-relevant concerns were (i) novelty clarity, i.e., whether the method goes beyond applying CTD/consistency distillation with an added reward loss (LYwS, 7SAP), (ii) baseline completeness and fairness, including the need for stronger one-step/flow comparisons and questions about optimized denoising-step settings (LYwS, 7SAP), and (iii) contribution framing, specifically that the reported speedups are largely inherited from CTM/CTD and should be attributed accordingly rather than framed as a new acceleration contribution (ixyB).

**Reviewer Concerns:**

The authors clarified the novelty as where and how reward is coupled to consistency distillation—student-only, clean single-step reward loss, decoupled from teacher training and actor–critic optimization—and showed that gains are not due to vanilla CTD alone (LYwS, ixyB). They strengthened comparisons by adding a one-step actor–critic baseline (Flow Q-Learning) and expanding CTD-vs-RACTD results across MuJoCo and Kitchen, with consistent gains on suboptimal data (LYwS, 7SAP, ixyB). The paper was also reframed to explicitly attribute acceleration to CTM/CTD and to emphasize the reward-aware adaptation as the main contribution (ixyB).

Remaining issues include concerns about incremental novelty relative to prior one-step/distillation methods, limited theory depth largely in the appendix (7SAP), and preferences for more standardized step-count comparisons (LYwS).

**Reviewer Scores:**

LYwS: Likely stays at 6, with higher confidence after added one-step baselines and clearer attribution of gains to the reward-aware design.

7SAP: Likely increase a bit, though novelty and theory concerns may remain.

ixyB: Likely stays at 6, with stronger acceptance support after corrected CTM/CTD attribution and broader ablations.

---

### Decision · Program_Chairs · 2026-01-26

Accept (Poster)